**Data Availability Statement:** The datasets generated during and analysed during the current

# "*If I've got latent TB, I would like to get rid of it*": Derivation of the CARD (Constraints, Actions, Risks, and Desires) Framework informed by South African healthcare worker perspectives on latent tuberculosis treatment

Ruvandhi R. Nathavitharana[1]*, Ananja van der Westhuizen[2]*, Helene-Mari van der Westhuizen[3], Hridesh Mishra[4], Annalean Sampson[4], Jack Meintjes[5], Edward Nardell[6], Andrew McDowell[7]°, Grant Theron[4]°

1 Beth Israel Deaconess Medical Center/Harvard Medical School, Division of Infectious Diseases, Boston, MA, United States of America, 2 Stellenbosch University, Department of Medicine, Cape Town, South Africa, 3 Department of Primary Care Health Sciences, University of Oxford, Oxford, United Kingdom, 4 DSI-NRF Centre of Excellence for Biomedical Tuberculosis Research, South African Medical Research Council Centre for Tuberculosis Research, Division of Molecular Biology and Human Genetics, Stellenbosch University, Cape Town, South Africa, 5 Unit for Infection Prevention and Control, Stellenbosch University and Tygerberg Hospital, Cape Town, South Africa, 6 Brigham & Women's Hospital, Division of Global Health Equity, Boston, MA, United States of America, 7 Tulane University, Department of Anthropology, New Orleans, LA, United States of America

° These authors contributed equally to this work.
* rnathavi@bidmc.harvard.edu (RRN); ananjavdw@gmail.com (AvdW)

## Abstract

### Background

Healthcare workers (HWs) have at least twice the risk of tuberculosis (TB) compared to the general population. There is growing emphasis on latent TB infection (LTBI) in high-risk populations. Yet we know little about HWs' perspectives of LTBI testing and treatment to inform implementation in high-incidence settings. We developed a qualitative networked approach to analyze HWs' perspectives on LTBI testing and treatment.

### Methods

We conducted 22 in-depth interviews with nurse and physician stakeholders, who had been recruited as part of a larger study evaluating TB transmission risk in HWs at Tygerberg Hospital, Cape Town, South Africa. We performed open coding to identify emergent themes and selective coding to identify relevant text citations. We used thematic analysis to inductively derive the CARD (Constraints, Actions, Risks, Desires) framework.

### Results

All HWs desired to avoid developing TB but few felt this was actionable. Despite LTBI knowledge gaps, safety and cost concerns, most HWs reported hypothetical willingness to take LTBI treatment. The CARD framework showed that desire and action related to LTBI testing

study are not publicly available since data consists of interview transcripts, which pose the risk that a person could be recognized from their interview narrative and the description of their facility. Any specific data requests can be made to the corresponding author, Dr. Nathavitharana, and/or the corresponding research ethics committee. This study was approved by the Stellenbosch University Health Research Ethics Committee, South Africa (Ref # N17/01/004, contact afortuin@sun.ac.za) and the Institutional Review Board of Brigham and Women's Hospital, Boston, USA (Protocol #: 2017P000539, contact agargiulo1@bwh.harvard.edu).

**Funding:** This research was funded by Harvard Medical School, Center for Global Health Delivery–Dubai. RRN is supported by National Institutes of Health Career Development Award (NIAID K23 AI132648-03) and an American Society of Tropical Medicine and Hygiene Burroughs Wellcome Fellowship. The funders had no role in study design, data collection and analysis, decision to publish, or preparation of the manuscript.

**Competing interests:** The authors have declared that no competing interests exist.

and treatment was clearly framed by the interactions between constraints, administrative action, and risk. The surprise HWs described on receiving a negative LTBI (Quantiferon-Plus) result suggests LTBI testing may recalibrate HWs' perceptions regarding the futility of actions to reduce their TB risk.

## Conclusions

LTBI testing and treatment are acceptable to HWs and could counteract the perceived inevitability of occupational TB infection that currently may limit risk reduction action. This should be coupled with administrative leadership and infrastructural support. The CARD analytic framework is a helpful tool for implementation scientists to understand current practices within complex health systems. Application of CARD could facilitate the development of contextually-relevant interventions to address important public health problems such as occupational TB.

## Background

Tuberculosis (TB) is a leading infectious cause of death in South Africa and globally [1]. It is estimated that 1.7 billion individuals worldwide (one quarter of the global population) have latent TB infection (LTBI) [2], which carries risk of progressing to active TB. The development of active TB can be prevented by administering TB preventive therapy (TPT) to people with LTBI [3, 4]. Yet, TPT is poorly implemented even in groups with high risk, including people living with HIV (PLHIV) and child contacts of a person with infectious TB [1].

The World Health Organization (WHO) estimates healthcare workers (HWs) have at least twice the risk of active TB compared to the general population [1], although the accuracy of these estimates is limited due to the lack of systematic reporting. Recommended screening for active TB in HWs is often not performed since occupational health systems in high-incidence countries are overburdened or absent [5, 6]. Specific challenges include the lack of trained occupational medicine personnel and concerns about confidentiality [5]. Internal and external stigma experienced by HWs affected by TB is a major care seeking barrier [7–9], especially if that care is offered through occupational mechanisms [10, 11]. Systematic reviews demonstrate high rates of LTBI in HWs in low and middle-income countries [12, 13], such as South Africa [14–16]. Since testing and treatment for LTBI in HWs is not currently implemented in high-incidence countries such as South Africa, the exact burden of LTBI in HWs is unknown.

Systematic review data, predominantly from low-incidence settings, indicates large drop-offs in the LTBI care cascade (including for health professionals), starting from the implementation of testing to uptake and completion of treatment [17]. Recent trials demonstrate the efficacy and safety of shor444ter TPT regimens [18–21], which have been incorporated into guidelines [22]. Although there is a renewed focus on scaling up TPT as part of elimination efforts [23], little is known about the willingness and barriers faced to access LTBI testing and TPT by high-risk asymptomatic populations in high-incidence settings like HWs. A pilot TPT implementation program in Swaziland reported that only 21% of eligible HWs started TPT, with barriers to acceptance including limited understanding of how TPT reduces TB risk, side effects, and access to medication (isoniazid in this study) [24]. Given the growing emphasis on LTBI treatment in high-risk populations that include HWs [22], there is an urgent need to

understand HW perspectives and practices on LTBI testing and treatment to develop and implement guidelines.

Studies examining HW clinical practice report a considerable gap between knowledge of a best practice and actually doing it, known as the "know-do gap" [25, 26]. Knowledge, Attitudes, and Practices (KAP) surveys and explanatory model interviews are typically the most prominent frames used to examine HWs' actions and roles in health systems [27–31]. KAP and other explanatory model studies assume reported belief or knowledge always guide clinical action but, like cognitive theories to understand individual behaviors such as the health belief model [32] and theory of planned behaviour [33], they present limited insights into the contextual background and dynamic effects of structures like economic and health system factors on individual behaviors [34–37].

A recent systematic review examining barriers and facilitators to TB infection control practices demonstrates that existing data focus on documenting poor TB infection control practices at the facility level and frequently frame the problem as being due to poor implementation of recommended guidelines by HWs, without consideration of the complex contextual factors including the lack of integrated occupational health efforts [38]. There is a critical need for analytic frameworks that capture interactions between individuals and the complex systems, in this case health facilities, in which they function to guide the design of interventions to reduce TB risk. A networked approach to qualitative research has been recommended to enable a deeper understanding of the linkages and multiple interactions between networked elements of TB care [37], which can be used to develop these analytic frameworks.

We conducted in-depth interviews with South African HWs in an observational study evaluating LTBI incidence. This qualitative research was done to understand linkages between drivers of HWs' perspectives and their resultant behavior associated with occupational LTBI testing and treatment. Our overall aim was to develop an analytic framework to understand how HWs perceive occupational TB risk and how mitigating strategies such as TPT could ultimately inform intervention design.

## Methods

### SAFE study context and setting

The SAFE study is a prospective observational cohort study evaluating the relationship between shared (rebreathed) air fraction estimate (SAFE) calculated using $CO_2$ level monitoring and the development of LTBI in HWs (medical and nursing staff) employed at Tygerberg Hospital, an 1800-bed tertiary hospital in Cape Town, South Africa that serves as a teaching site for Stellenbosch University.

### Sampling and Recruitment

Our 22 interview participants were a subset of the 187 HWs in the parent SAFE study. For the parent study, medical and nursing staff were approached for participation using convenience sampling by our research nurse, who presented the study to groups of eligible HWs after departmental or ward meetings. For this qualitative study, participants were purposively sampled to achieve systematic representation of different HW professions across a range of departments and to have 50% of participants with a positive baseline LTBI result (based on Quantiferon-Plus testing) that was known to both participant and researcher. To achieve this, our purposive sampling strategy involved dividing participants into five departments: Medicine, Surgery, Obstetrics and Gynaecology, Paediatrics and Emergency and sub-dividing by job description: junior doctor (six), senior doctor (six), junior nurse (five) and senior nurse (five). We used a random number generator (www.randomizer.org) to select subgroup

participants, who would be contacted via telephone. If a participant was unavailable (n = 12: one refused, three were on maternity leave, two were no longer at Tygerberg Hospital, one did not have time, one did not wish interview to be recorded and four did not respond), the next randomly assigned participant for that subgroup was approached. 68% were female. We continued interviews until we reached thematic saturation based on iterative real-time data review.

## Ethics

The study was approved by the Human Research Ethics Committee of the Faculty of Health Sciences at Stellenbosch University and Tygerberg Hospital, Cape Town, South Africa and the Institutional Review Board of the Brigham and Women's Hospital in Boston, USA. Written informed consent was obtained.

## Data collection and preparation

In-depth individual in-person interviews consisted of open-ended questions designed to elicit detailed knowledge of LTBI testing and treatment, and TB transmission. Questions addressed perceptions of personal occupational TB risk, individual and systems or infrastructural challenges related to TB transmission, knowledge of LTBI, and HW LTBI testing and treatment. We piloted our interview guide with two research nurses and made revisions to ensure our questions were clear and simply phrased. Interviews were conducted by a trained research team member (AvdW), who was a female medical student familiar with Tygerberg Hospital but was unfamiliar to participants. This combination of insider and outsider perspectives may have helped the interviewer to create rapport with participants. The interviews were conducted in English (spoken by all participants) in private locations on hospital premises between July 2nd-26th 2018. Participants did not receive reimbursement or incentives. Interviews averaged 45 minutes and were audio-recorded with permission. Field notes were used to contextualise findings. After each interview, a transcript was obtained using a professional transcriptionist and quality reviewed and corrected (AvdW, RRN). Transcripts were not returned to participants for review. Interview audio-recordings and transcripts were reviewed iteratively and adaptations were accordingly made to our interview guide (AvdW, RRN, AJM).

## Data management

Using an inductive approach informed by thematic analysis methods [39], two coders (AvdW, RRN) performed open coding and recoding to derive emergent themes using NVivo Qualitative Data Analysis Software Version 12 (QSR International). Selected codes were used to identify relevant interview quotes. Differences in coding were resolved through discussion. Themes were assembled into an explanatory analytic framework to understand challenges and opportunities for HW LTBI testing and treatment.

## Data analysis plan and Development of the TB-CARD framework

When planning our analysis, we wanted to develop a dynamic framework that analyzed interactions between HW perceptions and behaviors within the complex hospital ecosystem. Our initial thematic analysis was inductive. All interview transcripts were coded to identify key recurring themes from within the data. Interrater reliability was maintained by having the data dually coded by two independent reviewers (RRN and AvdW), with themes reviewed by a third reviewer (AMcD). Four main themes were initially identified. They were: risk perception

regarding occupational TB, perceptions related to HWs' duty of care to patients, TB infection control, and individual versus hospital responsibility for HW protection.

We then used a networked approach to consider different roles, resources and health systems factors related to occupational TB risk and risk reduction [37]. Using risk, care, information, and protection as categories to analyze and organize the themes a second time, we described four revised domains that included 1) individual and systemic constraints and 2) actions that HWs and health systems took in anticipation and as a result of 3) HWs' risk of occupational TB. Examining all references to constraint and action in a final level of data analysis we noted that a final domain, 4) desire, helped integrate the previous iterations of analysis. In this category we included the ways HWs talked about their desires as related to risk, action, care, and responsibility. Finally, we combined overlapping themes to identify four central domains related to occupational TB risk reduction by HWs, Constraints, Actions, Risks, Desires, or CARD.

The CARD domains were identified based on our inductive data analysis process. They provide a dynamic, multifactorial analytical frame with which to systematically and represent the dynamic relationship between these broader structures (constraints, risk) with the hospital and interpret HW behavior (actions, desire). Combining individual and systemic factors in a single analytical frame, the CARD framework schematizes and describes the interaction of parts within the networked ecosystem in which HWs function to contextualize and interpret their perspectives about and behavior around LTBI infection and treatment. The authors conducting the analysis are either clinician researchers or anthropologists who are well versed in biomedical data demonstrating the benefits of TPT. Though we worked to bracket these frames, they may have influenced our methods and analysis.

### COREQ assessment

We ensured that our manuscript includes the components listed in the Consolidated criteria for reporting qualitative research (COREQ) checklist [40].

## Results

We present our key findings here organized by the domains of the CARD framework (also see **Fig 1** which summarizes domains and sub-domains). See **Table 1** for a description of how the themes developed through thematic analysis informed this framework.

### CONSTRAINTS in LTBI knowledge, testing and treatment, and TB infection control

Although most HWs had some understanding of LTBI, responses identified knowledge gaps that could constrain future demand and uptake of LTBI testing and treatment.

*Latent TB means you do have TB, but it's in a dormant, latent phase. So you're not actually infected yet, but if your immune system would somehow become compromised, it could mean that that latent TB that's lying there dormant becomes active TB.*

Participant 82: junior nurse, medicine

This HW, along with others, mentioned that they thought progression to active TB generally occurred due to the failure of the immune system to control it, although they acknowledged that people without immune deficiencies could develop TB. This perception may limit uptake of LTBI to only those at highest risk such as PLHIV.

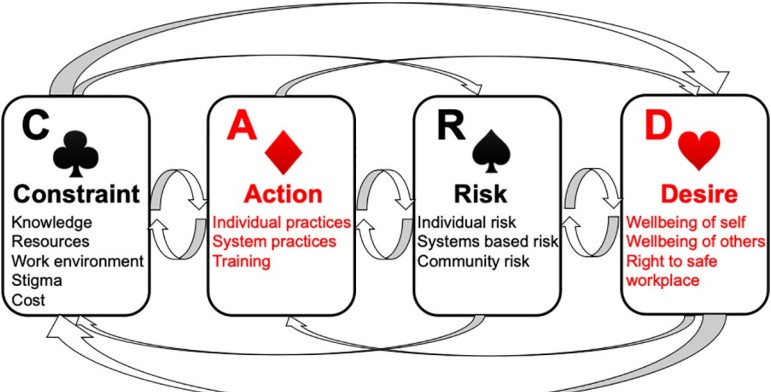

**Fig 1. Playing your CARDs right: The CARD framework can be considered as a deck of cards where ultimately outcomes (based on a combination of these cards), in this case related to the implementation of LTBI testing and treatment, are driven by the interactions between the major domains (Constraint, Action, Risk, Desire) and more specifically by the sub-domains (which may be adapted based on context, examples of which are shown on each card).**

Most participants were unsure if LTBI could be treated or thought that this should only be done in child contacts of someone with infectious TB.

*[re: LTBI treatment] a prophylactic with isoniazid is given, but it carries a risk depending how long [it is taken] and can cause side effects.*

Participant 4: senior doctor, medicine

When questions about LTBI treatment were posed, HWs referred to six month isoniazid regimens and none mentioned recently endorsed shorter regimens. This exclusive focus on isoniazid influenced health worker perceptions of what LTBI treatment would entail, making it a less attractive option.

Several doctors cited cost, from the perspective of the hospital, as a potential barrier to LTBI testing and treatment.

*Let's say hypothetically you treat it, you don't have latent TB anymore, but then you are still always exposed so, you are bound to get it again. So yes, it's beneficial treating it, but then how are you going to prevent exposure, which is almost everywhere now? I am just thinking that we are already squeezed in terms of our finances. Sometimes it's not an easy decision.*

Participant 144: senior doctor, paediatrics

Concerns about cost were raised in the context of the risk of re-exposure leading to re-infection, which was perceived to be high. This could affect perceptions about the efficacy of LTBI treatment and trepidation for an endless loop of treatment and re-exposure.

HWs reported their ability to implement TB infection control was constrained by the nature of their work, technological and hospital infrastructure, education, and cost. Several HWs reported prioritizing patients' urgent care needs often constrained implementation of infection control measures, predominantly relating to personal protective equipment (PPE) use.

*It's quite difficult to wear a mask all the time, because a lot of the masks are not breathable. They are very hot, sweaty. I feel it creates a barrier with the patient and that there is some sort*

**Table 1. Use of the CARD analytical framework to design and implement interventions intended to address complex health problems such as occupational TB (in this case, from both HW and health systems perspectives).**

| CARD domains and sub-domains | Description of identified themes | Example components of a multi-pronged intervention informed by CARD to address occupational TB |
|---|---|---|
| **CONSTRAINTS** | | |
| Knowledge | Limited understanding of LTBI process and risk, testing, and treatment. | Education of HWs and hospital leadership. |
| | Training gaps regarding TB-IC practices. | Reassessment of mode, timing, and frequency for training by facility team, based on iterative input from HW stakeholders. |
| Resources | Lack of capacity to implement occupational testing and treatment. | Resources (to include personnel and infrastructure) for staff to enable high quality occupational TB care and TB-IC. |
| | Lack of isolation/ventilation, triage and laboratory infrastructure. | Reassessment and re-design of TB-IC systems, requiring collaboration between administrators, leadership, HWs. |
| Work environment | Facility environment constraints may lead HWs to believe infection is unavoidable. | Change in health facility culture of practice will require buy-in from administrative and clinical leadership to provide support (also see resources) for LTBI testing and treatment, and TB-IC implementation. |
| | Duty of care felt by HWs may compromise decisions to use PPE or take other protective action. | Senior clinicians should model PPE use and other actions to protect wellbeing of self, staff, students, and patients, and should not promote attitudes that all HWs will be infected. |
| Stigma | TB stigma may impact PPE adherence and HWs' willingness to engage in care. | Training of HWs should include efforts to destigmatize TB and to engage HWs as TB champions. |
| Cost | HW awareness of health systems budget constraints | Buy-in from administrative leadership required, to be incentivized by the prospect of decreasing staff illnesses and aided by HW advocacy. |
| **ACTION** | | |
| Individual practices | Some HWs sought care for positive LTBI results but treatment was not recommended by clinicians they saw. | Wider training for clinicians seeing people with LTBI (and buy-in) regarding benefits of LTBI testing and treatment. |
| | HWs may seek own testing. | Need for facility based occupational health programs to address TB risk. |
| | TB-IC is underprioritized at the individual HW level. | Hs engagement through education and training. Emphasis on TB champions to model behavior may incentivize desired actions. |
| System practices | Occupational TB testing and treatment is not performed at the healthcare facility level. | Buy-in from administrative leadership and policy stakeholders will be needed to build or strengthen existing occupational health systems that should incorporate TB testing and treatment, including for LTBI. |
| Training | Physicians unaware of LTBI treatment benefits besides PLHIV and child contacts | Training of different groups of physicians, starting with occupational health physicians, to ensure understanding of benefit of testing and treatment. |
| **RISK** | | |
| Individual risk | High risk of occupational TB for HWs. | Specific HW education and training regarding LTBI testing and treatment to understand and mitigate risk. |
| | | Reframing of TB-IC training to develop HW TB champions focused on TB prevention (needs systems based support). |
| | | Engagement of senior clinicians who interact with junior staff (including students) to avoid efforts seeming futile. |
| Systems based risk | High risk of TB transmission in healthcare facility. | Reassessment of TB transmission risk, current TB-IC implementation gaps, and potential to re-design spaces or maximize existing ventilation. This will require active collaboration between administrative leadership and TB-IC team, and input from HWs should be sought. |
| Community risk | High risk of TB due to community transmission in high incidence country. | Requires a broader multidisciplinary approach with national guideline changes and increased funding to implement transmission prevention. |
| **DESIRE** | | |
| Wellbeing- self | HWs want to avoid becoming sick with TB. Most expressed hypothetical willingness for LTBI testing and treatment. | HWs need to be supported by the facilities employing them to be able to undertake testing for active and latent TB and receive treatment. This requires strengthening occupational health systems, along with concurrent efforts to improve TB-IC to prevent re-exposure. |
| Wellbeing- others | HWs expressed concern about transmission of TB to patients. | Integration of LTBI testing and treatment and TB-IC at systems-level, requiring active collaboration between administrative leadership with occupational health and TB-IC teams, and HW input. |

*(Continued)*

**Table 1.** (Continued)

| CARD domains and sub-domains | Description of identified themes | Example components of a multi-pronged intervention informed by CARD to address occupational TB |
|---|---|---|
| Right to safe workplace | HWs wished to work in a safe environment and acknowledged the health system's responsibility to provide this, including testing and treatment. | Require buy-in from administrative leadership to provide support for strengthened occupational health and TB-IC programs, incentivized by the prospect of decreasing staff illnesses and be aided by HW advocacy. |

Footnote: tuberculosis (TB), healthcare worker (HW), latent TB infection (LTBI), TB infection control (TB-IC).

*of stigma attached to it. I honestly don't like wearing a mask, but for my own safety, it would be better to wear a mask sometimes.*

Participant 154: junior doctor, surgery

Many HWs reflected on the challenges of wearing either a mask or a N95 respirator for prolonged periods. While these included physical discomfort for the HWs themselves, several raised concerns about the potential negative impact of anticipated stigma on the clinician-patient relationship.

The majority of HWs reported concerns about ventilation in the hospital due to poor air circulation and the inability to open windows. Many expressed the need for better isolation facilities with functioning exhaust fans to maintain negative pressure.

*There are only two rooms in the emergency department that are isolation rooms, but the rest are not negative pressure. So, patients, for example on [the medical admissions unit] where windows are closed, are next to each other and all of them are coughing. So, the ventilation system is poor, and that's a good environment for acquiring TB for the patients as well as the healthcare staff.*

Participant 4: senior doctor, medicine

This description of the environment that HWs work in and patients receive care in demonstrates the physical constraints that limit the effective implementation of TB infection control. This can lead to individual HW use of PPE often being perceived as the only infection control tool potentially available to create a safer health care environment.

### ACTION taken by HWs to reduce TB risk, including the impact of LTBI test results on reported behavior

HWs with a negative LTBI result expressed surprise given their perceived high risk of TB. In contrast, HWs with a positive LTBI result were not surprised but did express concerns about the implications of this result.

*I'm negative. I didn't expect that. I actually spoke to my other intern friends, and all of us thought we all probably have latent TB, because we work with a lot of TB patients. Although the threat of active TB is somewhat subdued, it hasn't changed the personal steps that I would take.*

Participant 159: junior doctor, Obstetrics and Gynaecology

*If an ambulance comes, I am ready with a mask. Earlier I would have gone to the patient without a mask, but now, because of this latent TB, I will be so cautious.*

Participant 71: senior nurse, Obstetrics and Gynaecology (in response to question about her reaction to receiving a positive LTBI result)

Most HWs (including those that had a positive LTBI result) reported that knowing their LTBI status motivated them to try to continue to reduce their risk although others reported that they would not change behaviour based on their results.

Nurses with a positive test were more likely than doctors to report seeking care based on a positive LTBI result.

*One of those ladies told me that her results came back that she has TB, but it's inactive, because her immune system is still strong. She went to the doctor [who] said there is no need to treat the TB because the TB is not active.*

Participant 81: senior nurse, medicine

When HWs sought care based on their positive LTBI test result, the physician care providers they consulted did not recommend TPT. This is likely reflective of the current guidelines in South Africa that focus on TPT for PLHIV and household contacts but may also reflect limited understanding of the efficacy of TPT.

No participants reported prior testing for active TB through a health facility based occupational health program.

*I usually do regular checks, because I work in an environment that is very high risk. So, just like when I test for HIV, I also check myself for TB. I do x-rays. It is on my own cost.*

Participant 142: senior doctor, medicine (pulmonology)

Several HWs mentioned making their own decision to undertake private testing with chest radiographs once yearly, due to their concerns about their occupational TB risk.

When responding to questions about reducing the risk of TB transmission, HWs primarily focused on their own actions rather than actions at the facility level.

*If it looks like TB and sounds like TB, then I wear the mask,*

Participant 9: junior doctor, paediatrics

*So, what I prefer to do, although the Sisters and the people all hate it, I really like to have open doors.*

Participant 126: senior doctor, emergency

HWs reflected upon their own use of PPE and often indicated that they would wear this only around patients with either known TB or a high concern for TB. A few mentioned that they would try to improve ventilation by opening windows or doors but described that this was not an approach prioritized by all HWs due to environmental constraints.

## RISK calibration for occupational TB may be facilitated by LTBI testing

All HWs were aware of their individual risk of occupational TB and knew other HWs who had developed active TB.

*I am constantly thinking about it [TB]. I'm aware of it, and I am concerned that I might eventually get it.*

Participant 126: senior doctor, emergency

*I use public transport, a bus. People cough a lot, and you don't know who is sick, who has TB. So, I can get TB anytime, not only here in the ward. I think it's safer here, more than the outside. Because here, I protect myself, we know we are dealing with TB patients here, so we know what to do.*

Participant 81: senior Nurse, medicine (TB ward)

HWs' perspectives regarding their occupational TB risk centered on balancing similarities and differences in TB risk within their communities compared to the hospital. While doctors focused primarily on their occupational risk, nurses were more likely to express recognition of the risk posed to them by high rates of community transmission in South Africa. Many endorsed the view that TB exposure and infection is an inevitable, intrinsic risk of being a HW.

*I think healthcare workers have accepted the fact that there is a possibility all of us have [latent] TB anyway.*

Participant 91: junior doctor, paediatrics

*I'm probably not as concerned as I should be. I'm not concerned, to be honest. I probably should be. I've become very desensitized.*

Participant 132: senior doctor, paediatrics

While the risk of occupational TB was concerning to most HWs, some expressed that they had become desensitized. Given their concerns about their presumed inevitable risk of TB, HWs expressed surprise upon receiving negative LTBI test results and learning that other HWs were also negative.

*I found it [negative LTBI result] quite surprising, especially with the amount of TB that I do encounter, and basically what you have been told as a student working in the hospital: that you most likely have latent TB. Even amongst our intern group, we discussed it, and it was surprising how many people were negative.*

Participant 162: junior doctor, Obstetrics and Gynaecology

Receiving a negative LTBI test result was counter to their expectation based on the information they received during training and from colleagues. This suggests that LTBI testing may help calibrate perceptions regarding the futility of TB infection control efforts to reduce risk, by increasing HWs awareness that they may not have all been previously infected already.

The majority of HWs mentioned the high risk of transmission due to undiagnosed TB in settings in all areas where patients are seen.

*[Patients in wards other than the TB ward may] have TB without the nurses knowing, so it's easy to infect the others if they are not on treatment.*

Participant 82: junior Nurse, medicine (TB ward)

*On average it will take us 24 to 36 hours to diagnose a patient with TB here. So there is no day here when I don't think I am exposed to TB.*

Participant 3: senior doctor, medicine

Although several HWs mentioned delays in obtaining a patient's active TB diagnosis, none mentioned the importance of triage followed by rapid testing or the use of universal precautions such as risk-reduction strategies. This means that infection control efforts are largely directed towards patients once they are diagnosed with TB (who are typically less infectious once on effective treatment) rather than the identification of all potentially infectious patients.

## DESIRE to undergo LTBI testing and receive treatment

While HWs acknowledged limitations in their LTBI understanding, many expressed the desire to know more and to be protected.

*Everybody should actually get tested, even if it's for latent TB, because it's great to know that you don't carry it, although you must always take precautions to not contract TB.*

Participant 185: junior nurse, surgery

*If it's something that can be treated safely, then yes, but if it's just something for the sake of knowing then no, because we know about the budget constraints in our health system.*

Participant 162: junior doctor, obstetrics & gynaecology

Several HWs thought LTBI testing should be offered to staff given the potential benefits of HWs knowing whether they had been infected, which may lead to improved implementation of TB infection control. Others mentioned the importance of being able to provide effective treatment for a testing programme to be worthwhile.

The majority of HWs expressed hypothetical willingness to take TPT, including those who described being desensitized to their TB risk, although several raised concerns about the safety of treatment (with reference to isoniazid-based regimens).

*I want to be protected more. I want to protect myself more, because I don't want to have TB. One day my immune system may become weaker, that TB is going to be active. If there is something to destroy that bacteria, I would want it.*

Participant 81: senior Nurse, medicine (TB ward)

*If I've got [latent] TB, I would like to get rid of it. I would take [treatment]. I'm aware of the damage that [TB] can cause, especially if it's pulmonary TB. They just don't recover fully from the first time they had it. I definitely do not want TB, and I think it's quite a bad disease to have. Even if I'm not sick, the fact that I know I've got TB, I would want to get the treatment.*

Participant 126: senior doctor, emergency

Desire to prevent short- and long-term damage due to TB and avoid perceived long-term risks of reactivation were common reasons for a positive perception of LTBI testing and treatment. HWs also expressed a desire for more action at the hospital-level to decrease TB transmission.

*You expect to be safe in your work environment.*

Participant 181: junior doctor, paediatrics

*I think [the hospital] must do it [TB testing] for free for us. You see, because we work with the patients in their hospital. So you can get it [TB] at any time.*

Participant 17: junior nurse, emergency

*They should make provision for [active TB testing done by the hospital]. I think indirectly it will save costs, because if your workers are healthy and not sick, they will continue working, they will provide a better service.*

Participant 126: senior doctor, emergency

*We don't want to be walking around with active TB and you are trying to help others and in the mean time you are actually making them sick.*

Participant 154: junior doctor, surgery

Several HWs cited the concern of transmitting TB to patients as a reason for receiving HW testing. All agreed that active TB testing should be provided by the hospital given the increased occupational risk. Some mentioned long-term benefits to the hospital regarding reducing the impact of HW illness on staffing. This linked with health worker expectations of having a safe healthcare environment–both to work in–and for patients to receive care in. Occupational health services that provide active and latent TB testing and treatment were described as a pre-requisite for this.

## Discussion

Our study provides an in-depth examination of HWs' perspectives on LTBI testing and treat-ment, in the context of unusually high occupational risk in a high TB incidence setting and global impetus for LTBI treatment programs. We derived the novel CARD framework using inductive data analysis and a dynamic and social science informed network model of human behavior to analyze our qualitative data, with a view to informing implementation planning. We demonstrate that LTBI testing and treatment are acceptable and indeed desirable to HWs. Our results suggest HW TB testing and treatment could counteract the perceived inevitability of occupational TB infection that may currently limit risk reduction. We recommend occupa-tional LTBI testing and treatment programmes should be coupled with efforts to strengthen TB infection control, which will reduce HW re-exposure risk. Our CARD analysis highlights the central importance of administrative leadership and infrastructural support, rather than focusing only on HW behaviour change.

The CARD analytic framework can enable implementation scientists to systematically doc-ument the interactions between individual behaviors, social aspirations, knowledge and risks, in the context of the structural, infrastructural, and systemic barriers that shape them (see **Fig 1**). CARD analysis showed that HWs in our interview cohort had a strong desire to prevent themselves from developing TB. They demonstrated acting on this desire by undertaking their own active TB testing privately and seeking care based on the LTBI-positive results obtained through the parent study. CARD also revealed that the majority of HWs in our interview cohort were potentially willing to take TPT if it was recommended and safe. Several HWs cited concerns about their potential risk to patients should they develop TB as a rationale for testing and treatment. This duty of care felt by HWs towards patients may be a source of motivation to emphasize as part of behavior change strategies. Many HWs expressed resignation regard-ing their perceived intrinsic and inevitable risk of acquiring TB. HWs who discovered that they were not, in fact, infected at baseline were surprised. This suggests that a LTBI testing pro-gram may calibrate HWs' risk perceptions and provide a powerful argument against the futility of efforts to decrease risk (such efforts could include TPT and system-wide TB infection con-trol efforts).

A key strength of this tool, unlike KAP surveys, is that CARD can capture *desire* and per-ception of *risk* as well as reported *action* and *constraints*; thereby informing interventions that are more likely to improve health outcomes without stigmatizing HW or blaming implementa-tion challenges on their behavior. We provide an example of how themes from inductive the-matic analysis can be categorized into CARD domains and applied to develop actionable strategies based on context-specific data, at the individual and health system level, that can inform the design and implementation of interventions (**Table 1**). The CARD framework also highlighted important potential barriers to LTBI testing and treatment. HWs expressed con-cerns about side effects of LTBI treatment, which may have been impacted by seeing patients experience side effects of TB drugs. Knowledge of shorter regimens with better safety profiles was also limited. HWs' desire to reduce their TB risk and willingness to consider TPT was tem-pered by the broader view of the high risk of re-exposure given current workplace constraints, which many HWs were concerned could compromise long-term treatment efficacy. When HWs sought medical evaluation after obtaining a positive LTBI test result, their physician care providers did not recommend LTBI treatment. This may reflect a training gap for physicians in high incidence settings regarding the perceived efficacy of LTBI treatment and perception that this should only be offered to people with HIV or child contacts. Although people with LTBI have a significantly lower risk (79%) of progressive TB after reinfection than those who are initially uninfected [41], we are not aware of evidence that suggests LTBI treatment may mitigate this lower risk. TPT has demonstrated a durable effect even in medium and high inci-dence settings [4, 42, 43]. Several large trials, that included sites in high TB incidence countries, have determined the efficacy and safety of shorter TPT regimens [18, 19, 21]. Use of the CARD framework can thus strengthen the design of evidence based interventions, such as LTBI testing and treatment, by identifying potential barriers (constraints, actions, risks) and enablers (actions, risks, desires). We emphasize that CARD can illuminate the overlap between potential intervention components and highlight the need for multi-pronged approaches to address this type of complex global health problem.

Several doctors mentioned budgetary constraints as a barrier to HW LTBI testing. This sug-gests their training exposed them to the lack of adequate funding for high-quality healthcare, with the implication that they appeared to prioritize the potential short-term economic cost to the health system over their personal health. Yet, overwhelmingly the HWs interviewed acknowledged the need for the hospital system to do more to create safe environments for both themselves and patients. Reprioritizing the importance of managerial and administrative mea-sures, which are at the top of the TB infection control hierarchy yet were rarely mentioned by HWs, remains a challenge for TB infection control implementation efforts. We acknowledge the challenges of implementing LTBI testing and treatment in real world hospital settings in high incidence countries, where resources for implementing other TB infection control mea-sures such as expanded triage, airborne isolation capacity, and ventilation, are limited. Nonethe-less, we think there is an imperative to reduce the unacceptable risk of occupational TB faced by HWs [44], which can be mitigated by comprehensive approaches that include LTBI testing and treatment [24]. TB transmission prevention interventions should not focus on changing HW actions alone but must be coupled with administrative and infrastructural support. This could facilitate implementation of recommended TB infection control measures, in conjunction with LTBI testing and TPT as part of an occupational health program, which HWs thought was the responsibility of the hospital to provide. Adopting a multi-pronged approach to addressing HW TB is a necessary component of TB transmission prevention strategies. Incorporating TB infec-tion control within broader health systems strengthening efforts can further these goals [45].

It is important to note that HWs' TB risk in this setting was not simply occupational. HWs' understood their risk as high due to living in a high transmission setting. HWs' perception of

intrinsic risk likely constrains both individual and administrative action because of its inferred futility. Although our CARD analysis did not reveal major differences in risk perception based on HW profession, it was apparent that for predominantly nurses, risk reduction at the hospital is only one part of a broader urgently needed risk reduction strategy that includes community transmission prevention measures. In contrast, for doctors, risk reduction at the hospital would significantly lower exposure risk. This is reflective of differing individual socio-economic determinants that impact TB risk. This underscores the importance of innovative and comprehensive strategies to reduce TB risk in affected communities, which should similarly include testing and treatment for LTBI.

Our study was limited to doctors and nurses who chose to enroll in the parent study and may not be representative of all HWs in the academic hospital setting or indeed other settings. Research with HWs in primary care facilities and community health posts is also urgently needed, along with data that elicits perspectives from health system administrators. Since LTBI treatment was not offered as part of the study, HWs reported willingness for TPT was hypothetical. However, the intent of our analysis is to generate an account of perceptions and practices related to HW LTBI testing and treatment intent in a high incidence setting, which we think may have transferability to different practice settings. We hope that this will lead to an iterative process to use the CARD framework to inform data collection about implementation challenges such as the development of occupational health LTBI testing and treatment strategies in conjunction with improved TB transmission prevention measures. Of note, our study was performed prior to the COVID-19 pandemic. Understanding the potential impact of COVID-19 on practices related to TB infection control, including the use of PPE, will be important for intervention design. We propose that our study findings, including the development of the CARD framework, may provide critical information to support broader implementation of airborne infection control measures and begin to create a dynamic, analytical approach to qualitative research on interventions such as LTBI testing and treatment.

## Conclusions

Derivation and application of CARD enabled us to understand the desire of HWs in a high incidence setting in a manner that informs their actions to reduce occupational TB high risk in the context of individual- and system-level constraints. Our data suggest that LTBI testing and treatment are acceptable to HWs in high incidence settings and could help to counteract the perceived inevitability of occupational TB infection that currently limits risk reduction action. This must be coupled with efforts to decrease re-exposure through administrative leadership at the facility level to provide infrastructural support for TB infection control implementation and national leadership to decrease TB risk in communities. We propose the CARD analytic framework can be used by implementation scientists to facilitate the development of contextually relevant interventions to address complex global health problems, such as occupational TB.

## Supporting information

**S1 File.**
(DOCX)

## Author Contributions

**Conceptualization:** Ruvandhi R. Nathavitharana, Ananja van der Westhuizen, Jack Meintjes, Edward Nardell, Grant Theron.

**Data curation:** Ruvandhi R. Nathavitharana, Ananja van der Westhuizen, Helene-Mari van der Westhuizen, Grant Theron.

**Formal analysis:** Ruvandhi R. Nathavitharana, Ananja van der Westhuizen, Helene-Mari van der Westhuizen, Andrew McDowell.

**Funding acquisition:** Ruvandhi R. Nathavitharana, Edward Nardell, Grant Theron.

**Investigation:** Ruvandhi R. Nathavitharana, Ananja van der Westhuizen, Hridesh Mishra, Annalean Sampson, Edward Nardell, Grant Theron.

**Methodology:** Ruvandhi R. Nathavitharana, Ananja van der Westhuizen, Andrew McDowell, Grant Theron.

**Project administration:** Ruvandhi R. Nathavitharana, Hridesh Mishra, Annalean Sampson, Jack Meintjes, Edward Nardell, Grant Theron.

**Resources:** Ruvandhi R. Nathavitharana, Jack Meintjes, Edward Nardell, Grant Theron.

**Software:** Ruvandhi R. Nathavitharana.

**Supervision:** Ruvandhi R. Nathavitharana, Edward Nardell, Grant Theron.

**Validation:** Ruvandhi R. Nathavitharana, Andrew McDowell, Grant Theron.

**Visualization:** Ruvandhi R. Nathavitharana, Grant Theron.

**Writing – original draft:** Ruvandhi R. Nathavitharana, Ananja van der Westhuizen, Helene-Mari van der Westhuizen, Hridesh Mishra, Annalean Sampson, Jack Meintjes, Edward Nardell, Andrew McDowell, Grant Theron.

**Writing – review & editing:** Ruvandhi R. Nathavitharana, Ananja van der Westhuizen, Helene-Mari van der Westhuizen, Hridesh Mishra, Annalean Sampson, Jack Meintjes, Edward Nardell, Andrew McDowell, Grant Theron.

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
