## [Decision Letter · Decision Letter 0]

18 Mar 2021

PONE-D-21-03741

“If I've got latent TB, I would like to get rid of it”: Derivation of the CARD (Constraints, Actions, Risks, and Desires) Framework informed by South African healthcare worker perspectives on latent tuberculosis treatment

PLOS ONE

Dear Dr. Nathavitharana,

Thank you for submitting your manuscript to PLOS ONE. After careful consideration, we feel that it has merit but does not fully meet PLOS ONE’s publication criteria as it currently stands. Therefore, we invite you to submit a revised version of the manuscript that addresses the points raised during the review process.

We look forward to receiving your revised manuscript.

Kind regards,

Amrita Daftary

Academic Editor

PLOS ONE

Journal Requirements:

2. Please include a copy of the interview guides used as a Supporting Information file.

4. Please amend the manuscript submission data (via Edit Submission) to include authors Ananja van der Westhuizen, Helene-Mari van der Westhuizen, Hridesh Mishra, Annalean Sampson, Jack Meintjes, Edward Nardell, Andrew McDowell, Grant Theron.

Reviewers' comments: reviewer 1 comments are in attachments. Reviewer 2 comments are pasted below. 

Reviewer's Responses to Questions

**Comments to the Author**

1. Is the manuscript technically sound, and do the data support the conclusions?

Reviewer #1: Yes

Reviewer #2: Partly

2. Has the statistical analysis been performed appropriately and rigorously? 

Reviewer #1: N/A

Reviewer #2: N/A

3. Have the authors made all data underlying the findings in their manuscript fully available?

Reviewer #1: No

Reviewer #2: Yes

4. Is the manuscript presented in an intelligible fashion and written in standard English?

Reviewer #1: Yes

Reviewer #2: Yes

5. Review Comments to the Author

Reviewer #1: Complete review has been attached

The author has not made the data underlying the findings fully available, due to participant narrations in transcripts potentially jeopardizing their anonymity.

The manuscript has been written in an intelligible fashion and in standard English, but some sentences require minor restructuring.

Reviewer #2: This qualitative research study looks at clinician perceptions of occupational health risk of TB infection (LTBI) and disease and the use of TB preventive therapy (TPT) at a teaching hospital in Cape Town, South Africa. The subject is of primary importance, as risk of TB infection and disease is high in this setting, and health workers are at twice the risk compared to the general population given their routine exposure. Based on 22 in-depth interviews with nurses and doctors with both positive and negative LTBI results across 5 hospital departments, the authors make a number of conclusions, most notably, that knowing LTBI status may provide the impetus for clinicians to take TPT and that system-level constraints may prevent this from coming to fruition. In addition, the authors present a unified framework that encourages the consideration of systemic barriers and facilitators in the uptake of interventions among health workers of use to future researchers. The manuscript is very well organised and written, and findings provide important evidence to advocate for routine TB screening and prophylaxis in high-burden clinical settings. There are some concerns with the certainty with which the authors present the findings and the presence of bias toward TPT implementation at all cost, which can be tempered with some additions and revisions to the findings and discussion sections.

Major suggestions are as follows:

The methods would benefit from additional information on the authors who conducted the analysis. It should be clear to readers that data is viewed from a biomedical perspective where TPT is of benefit.

I would caution the authors to shy away from definitive statements around the findings. For example “we demonstrate that LTBI testing and treatment are acceptable and desirable to HWs” (Line 442-443); “CARD analysis demonstrated that HWs have a strong desire to prevent themselves from developing TB” (line 453-4) and discussing what “the majority” of health workers reported with regard to TPT willingness in the study, as these statements misrepresent what can be concluded based on a very small set of data. Qualitative research is intended to provide a deeper understanding of a phenomenon, often from multiple perspectives, not to draw far reaching conclusions for the broader population from which the sample was drawn. The wording around these findings should be softened to reflect the study design. It can be helpful to intersperse limitations within the discussion, as relevant, rather than leaving these for a section at the end, as is typically done in quantitative research. It may also help to touch on the transferability of findings (as opposed to generalisability – line 492), so as not to miss the importance of what has been found.

The authors present a lot of rich data on barriers and facilitators, some of which is not adequately discussed. Table 1 contains a lot of information, some of which is not presented in text (e.g. stigma) and some for which solutions are oversimplified (e.g. senior physicians should model use of PPE). Rather than try to tackle all of these data in one paper/table, it may be more suitable to discuss the main findings (e.g. knowing LTBI status may improve TPT uptake among HWs) in-depth, while acknowledging the complexity of other factors, and how they might affect uptake. The authors must be cautious not to confuse complex concerns with “training gaps” or “gaps in latent TB understanding.” This displays a bias toward TPT-use without considering the genuine concerns and lived experience of clinicians in this setting.

For example:

As TB is endemic and comparatively harder to catch than COVID, it may not be realistic to expect doctors and nurses to wear PPE all the time, especially for those who are also at a high risk outside the hospital environment. It is important to acknowledge that there are real limitations to current infection control practices in this setting, and suggests we need better long-term solutions. This could make TPT even more attractive as an intervention at the institution level.

Re-exposure is also a reality in this setting. We don’t fully understand how TPT may affect innate immunity, and this concerns some of the clinical staff. It is important to acknowledge these concerns which suggest that not all HWs will opt for TPT until better evidence is available.

Similarly, side effects of the regimen are a concern, regardless of whether it is IPT, 3HP, or some other regimen. Nurses especially would be familiar with the hardships of side effects based on patient experience. Other research has suggested that this prevents health workers and patients from opting for prophylaxis, which may limit uptake.

It would also be helpful to readers from outside LMICs to provide some context around cost concerns. What does a TB ward look like? Are negative pressure rooms available? What other competing risks to patients and HWs are present? How might we go about advocating for systemic changes to prevent TB and other nosocomial infections in light of limited resources?

Minor comments:

Introduction:

There are a lot of acronyms introduced that may be unfamiliar to readers outside the field of TB. I would suggest spelling out infection control and Tygerberg hospital to reduce the number of acronyms.

Line 94: Please reword this sentence, as hypothesis testing is not appropriate for qualitative data, as the findings are inductively derived.

Methods:

Was a particular methodological orientation used in this study (e.g. ethnography, grounded theory)?

Line 144: Who was the interview guide pilot tested on, and were any revisions made? It would be helpful to share the interview guide in a supplement. In particular it would be interesting to note in the findings whether any of the interviewees had previously had TB or lost a colleague to TB.

Information on gender of participants should be included in methods or findings.

Lines 164: This section intersperses a lot of findings in the methods. Perhaps this detail would be better shared as a table or graph in the beginning of findings. I leave this to the authors to decide.

Line 167: It would be helpful to add a line or two explaining a network approach for those who are unfamiliar.

Lines 172-174: this sentence was a bit confusing and could benefit from rewording.

Lines 184-85: Is it relevant that these data were analysed in parallel with SAFE study results? Did one inform the other?

Findings:

Lines 374-377: This sentence is long and difficult to follow. Please consider rewording.

Discussion:

Lines 438-439: I would advise rewording this sentence, as this is not the first study to look at perceptions of TPT in HWs in a high-incidence setting; however, the findings are very relevant.

Lines 499-501: This is a very important finding and a strength of your approach.

6. PLOS authors have the option to publish the peer review history of their article (what does this mean?). If published, this will include your full peer review and any attached files.

Reviewer #1: No

Reviewer #2: **Yes: **Jody Boffa

---

## [Author Response · Author response to Decision Letter 0]

6 May 2021

PLOS ONE

“If I've got latent TB, I would like to get rid of it”: Derivation of the CARD (Constraints, Actions, Risks, and Desires) Framework informed by South African healthcare worker perspectives on latent tuberculosis treatment.

Thank you for the opportunity to read and review your manuscript. I particularly enjoyed reading the manuscript, as it is a qualitative study looking at latent TB, which is particularly relevant and a large problem in South Africa. Overall, the author has made compelling points from the perspective of Healthcare workers. However, some revisions are needed. Overall, it is recommended that the author goes back to the manuscript to rephrase or break up certain sentences that are long. Fixing the structure of the sentences will help the overall flow or provide clarity in their statements. It would also be helpful if the author gave some further clarity on some of the challenges or limitations with TB screening given the nature of the occupational health systems, as per the available literature, to further support this statement provided in the introduction. The author makes use of the COREQ guidelines, offering additional detail on the qualitative methodology used. The author also uses an analytical framework which guides and structures the analysis process and results. Thank you for using an innovative and creative approach (as per Figure 1 and Table 1), to address the gaps and issues as discussed in this manuscript. 

1.1 Thank you. We have revised the manuscript for clarity, striving for shorter sentence structure to improve readability.

Our responses to reviewer comments are in bold with revised text provided in italics. Line references refer to the manuscript version with tracked changes.

1.2 We have added a sentence about specific challenges for TB screening in line 58-59 including the lack of trained occupational medicine personnel and concerns about confidentiality.

Specific challenges include the lack of trained occupational medicine personnel and concerns about confidentiality.

Additional revision recommendations have been listed below;

ABSTRACT

1.3 Line 13: Consider changing starting the sentence with ‘we did’, as this currently sounds more informal, to ‘we conducted’.

We have made this revision.

1.4 Line 20-21: Rephrase; Despite LTBI knowledge gaps and safety and cost concerns’ to ‘Despite LTBI knowledge gaps, safety and cost concerns’ 

 We have made this revision.

INTRODUCTION

1.5 Line 88-89: This statement may come through stronger if more clarity was provided

We edited this statement for clarity:

There is a critical need for analytic frameworks that capture interactions between individuals and the complex systems, in this case health facilities, in which they function to guide the design of interventions to reduce TB risk.

1.6 Line 83-97: Suggestion would be to rephrase this sentence to provide further clarity

We revised this section for clarity:

We conducted in-depth interviews with South African HWs in an observational study evaluating LTBI incidence. This qualitative research was done to understand linkages between drivers of HWs’ perspectives and their resultant behavior associated with occupational LTBI testing and treatment. Our overall aim was to develop an analytic framework to understand how HWs perceive occupational TB risk and how mitigating strategies such as TPT could ultimately inform intervention design.

METHODS

1.7 Line 115-118: ‘For this qualitative study, participants were purposively sampled to achieve systematic representation of a variety of perspectives and have 50% of participants with a positive baseline LTBI result (based on Quantiferon-Plus testing) that was known to both participant and researcher’. Considering including and ‘to have 50%’ so the sentence flows better. 

We have made this revision.

1.8 As per the COREQ guidelines, could you provide more information on the interviewers training? Were they part of the parent study, or were they unfamiliar to the participants?

We have clarified in line 177 that the interviewer was unfamiliar to the participants.

Interviews were conducted by a trained research team member (AvdW), who was a female medical student familiar with Tygerberg Hospital but was unfamiliar to participants.

1.9 Line 151-154 needs rephrasing 

We have revised this text in lines 185-186 as follows:

Interview audio-recordings and transcripts were reviewed iteratively and adaptations were accordingly made to our interview guide (AvdW, RRN, AJM).

1.10 Line 170 – is missing a ‘,’ after context

We removed this clause as part of our revisions.

1.11 The author indicates, ‘To interpret HW perspectives in context our initial coding included four main themes: risk perception, regarding occupational TB, perceptions related to HWs’ duty of care to patients, TB-IC, and individual versus hospital responsibility for HW protection’, how were these themes derived? Could the author operationalize this process and how the themes were selected/emerged?

We have revised this section extensively in response to comments from both reviewers which we hope clarifies this question regarding the identification of themes, please see lines 197-241.

1.12 Line 164: Please introduce here that the section pertains to the initial analysis that guided the development of the CARD framework, as initially it seems like the author is already presenting some results, which are then later applied to the data for analysis.

We have revised this section extensively, which we hope clarifies that this is part of our methodology for developing the CARD framework, please see lines 198-247. 

1.13 As per the COREQ guidelines, were transcripts returned to participants for review? Did any participants receive a reimbursement or incentive? How was interrater reliability maintained or final presented themes evaluated amongst the team? 

We have added the following text in lines 180-181 and 184-185 respectively:

Participants did not receive reimbursement or incentives. 

Transcripts were not returned to participants for review.

We added the following text in lines 200-202 regarding interrater reliability:

Interrater reliability was maintained by having the data dually coded by two independent reviewers (RRN and AvdW), with themes reviewed by a third reviewer (AMcD).

RESULTS

1.14 The author presents a statement and a quote for each topic, under each theme. It would strengthen the manuscript, if the author offered some clarification or interpretation of the statements and quotations, to strengthen and further represent the sub-theme.

We have now made revisions throughout the results section such that there is text below and after the quotes that illustrate the themes identified from thematic analysis. This has enabled us to provide additional interpretation of the chosen quotes and statements as requested by the reviewer. We provide an example here:

Although most HWs had some understanding of LTBI, responses identified knowledge gaps that could constrain future demand and uptake of LTBI testing and treatment. 

Latent TB means you do have TB, but it’s in a dormant, latent phase. So you’re not actually infected yet, but if your immune system would somehow become compromised, it could mean that that latent TB that’s lying there dormant becomes active TB. 

Participant 82: junior nurse, medicine

This HW, along with others, mentioned that they thought progression to active TB generally occurred due to the failure of the immune system to control it, although they acknowledged that people without immune deficiencies could develop TB. This perception may limit uptake of LTBI to only those at highest risk such as PLHIV.

1.15 Typo: Line 250 missing a full-stop at the end

We have made this revision.

1.16 Could the author unpack or provide more clarity to the following statement: ‘Many reported discomfort, stigma, and potential negative impact on the clinician-patient relationship made it challenging to wear either a mask or a N95 respirator for prolonged periods’.

We have revised this statement in lines 391-394 for clarity:

Many HWs reflected on the challenges of wearing either a mask or a N95 respirator for prolonged periods. While these included physical discomfort for the HWs themselves, several raised concerns about the potential negative impact of stigma on the clinician-patient relationship.

1.17 Consider rephrasing line 289 – 291.

We have revised these statements in lines 455-456 and 463-464 as suggested:

No participants reported prior testing for active TB through a health facility based occupational health program. 

Several mentioned making their own decision to undertake private testing with chest radiographs once yearly, due to their concerns about their occupational TB risk. 

1.18 Line 374-377: Consider breaking into smaller sentences.

We have revised this section in lines 516-519 as suggested:

Several HWs thought LTBI testing should be offered to staff given the potential benefits of HWs knowing whether they had been infected, which may lead to improved implementation of TB infection control. Others mentioned the importance of being able to provide effective treatment for a testing programme to be worthwhile. 

DISCUSSION

1.19 Line 439 – 442: The ‘social science informed network model of human behavior’ was not mentioned in the methodology until in the discussion.

Thank you – we have now clarified this in the methodology in lines 219-220:

We then used a networked approach to consider different roles, resources and health systems factors related to occupational TB risk and risk reduction[37]. 

1.20 Line 460 – 462: Requires rephrasing 

We have revised these statements in lines 716-719 as suggested:

Many HWs expressed resignation regarding their perceived intrinsic and inevitable risk of acquiring TB. HWs who discovered that they were not, in fact, infected at baseline were surprised. 

1.21 Line 469 – 475: How are these conclusions regarding the tool derived from the data presented? 

We clarify in lines 726-729 that these are not conclusions but describe how themes identified from thematic analysis can be analyzed using the CARD framework, which enables intervention design to be informed by context-specific data and demonstrate an example of this in Table 1.

We provide an example of how themes from inductive thematic analysis can be categorized into CARD domains and applied to develop actionable strategies based on context-specific data, at the individual and health system level, that can inform the design and implementation of interventions (Table 1).

1.22 Line 488 – 492: This longer sentence needs to be broken up; lacks clarity

We have removed this sentence. 

Reviewer #2: This qualitative research study looks at clinician perceptions of occupational health risk of TB infection (LTBI) and disease and the use of TB preventive therapy (TPT) at a teaching hospital in Cape Town, South Africa. The subject is of primary importance, as risk of TB infection and disease is high in this setting, and health workers are at twice the risk compared to the general population given their routine exposure. Based on 22 in-depth interviews with nurses and doctors with both positive and negative LTBI results across 5 hospital departments, the authors make a number of conclusions, most notably, that knowing LTBI status may provide the impetus for clinicians to take TPT and that system-level constraints may prevent this from coming to fruition. In addition, the authors present a unified framework that encourages the consideration of systemic barriers and facilitators in the uptake of interventions among health workers of use to future researchers. The manuscript is very well organised and written, and findings provide important evidence to advocate for routine TB screening and prophylaxis in high-burden clinical settings. There are some concerns with the certainty with which the authors present the findings and the presence of bias toward TPT implementation at all cost, which can be tempered with some additions and revisions to the findings and discussion sections.

2.1 Thank you. We have softened the language throughout to address the concern about bias towards TPT implementation and acknowledged the real world challenges of addressing LTBI testing and treatment in HWs, given the constraints in high incidence countries. We have incorporated these suggestions as outlined in more detail below.

Major suggestions are as follows:

2.2 The methods would benefit from additional information on the authors who conducted the analysis. It should be clear to readers that data is viewed from a biomedical perspective where TPT is of benefit.

We have added the following statement in the Methods, lines 238-241 to address this point:

The authors conducting the analysis are either clinician researchers or anthropologists who are well versed in biomedical data demonstrating the benefits of TPT. Though we worked to bracket these frames, they may have influenced our methods and analysis.

2.3 I would caution the authors to shy away from definitive statements around the findings. For example “we demonstrate that LTBI testing and treatment are acceptable and desirable to HWs” (Line 442-443); “CARD analysis demonstrated that HWs have a strong desire to prevent themselves from developing TB” (line 453-4) and discussing what “the majority” of health workers reported with regard to TPT willingness in the study, as these statements misrepresent what can be concluded based on a very small set of data. Qualitative research is intended to provide a deeper understanding of a phenomenon, often from multiple perspectives, not to draw far reaching conclusions for the broader population from which the sample was drawn. The wording around these findings should be softened to reflect the study design. It can be helpful to intersperse limitations within the discussion, as relevant, rather than leaving these for a section at the end, as is typically done in quantitative research. It may also help to touch on the transferability of findings (as opposed to generalisability – line 492), so as not to miss the importance of what has been found.

Thank you. We have softened language throughout. We have also clarified that these statements refer to HWs in our interview cohort (lines 705 and 709) to avoid misrepresentation of these findings to HWs more widely based on limited data.

We have removed the sentence with the comment about generalizability. We think this comment still applies, since the population studied have a higher than average rate of reinfection which limits study generalizability. We appreciate the suggestions regarding transferability rather than generalizability of our findings and have added text accordingly in lines 859-862:

However, the intent of our analysis is to generate an account of perceptions and practices related to HW LTBI testing and treatment intent in a high incidence setting, which we think may have transferability to different practice settings.

2.4 The authors present a lot of rich data on barriers and facilitators, some of which is not adequately discussed. Table 1 contains a lot of information, some of which is not presented in text (e.g. stigma) and some for which solutions are oversimplified (e.g. senior physicians should model use of PPE). Rather than try to tackle all of these data in one paper/table, it may be more suitable to discuss the main findings (e.g. knowing LTBI status may improve TPT uptake among HWs) in-depth, while acknowledging the complexity of other factors, and how they might affect uptake. The authors must be cautious not to confuse complex concerns with “training gaps” or “gaps in latent TB understanding.” This displays a bias toward TPT-use without considering the genuine concerns and lived experience of clinicians in this setting.

Thank you. Table 1 is intended to be a comprehensive summary table that demonstrates example components of a multi-pronged intervention to address HW TB with a focus on LTBI testing and treatment, based on context-specific data. We think that this will help implementation scientists to consider how the CARD framework can be applied to develop contextually relevant approaches to complex implementation problems such as occupational TB. We sought to address the concerns about oversimplification of solutions by highlighting the need for multi-pronged approaches (lines 748-751) and emphasize that a combination of solutions that are not only focused on individual behavior change (lines 832-834) will be needed:

We emphasize that CARD can illuminate the overlap between potential intervention components and highlight the need for multi-pronged approaches to address this type of complex global health problem.

TB transmission prevention interventions should not focus on changing HW actions alone but must be coupled with administrative and infrastructural support.

We have revised Table 1 to ensure all its contents are appropriately discussed in the text without and distilled the information to avoid unnecessary duplication and improve clarity, while noting that the use of the CARD framework identified overlapping themes with potentially overlapping solutions. We also note that Reviewer 1 found Table 1 to be helpful and informative (comment 1.1).

We have also added language to acknowledge the genuine concerns and lived experiences of clinicians in this setting regarding the complexity of LTBI testing and treatment, please see response to 2.4.4.

For example:

2.4.1 As TB is endemic and comparatively harder to catch than COVID, it may not be realistic to expect doctors and nurses to wear PPE all the time, especially for those who are also at a high risk outside the hospital environment. It is important to acknowledge that there are real limitations to current infection control practices in this setting, and suggests we need better long-term solutions. This could make TPT even more attractive as an intervention at the institution level.

- We have revised text to address these points in lines 758-761 and 832-836:

Reprioritizing the importance of managerial and administrative measures, which are at the top of the TB infection control hierarchy yet were rarely mentioned by HWs, remains a challenge for TB infection control implementation efforts.

TB transmission prevention interventions should not focus on changing HW actions alone but must be coupled with administrative and infrastructural support, in conjunction with LTBI testing and TPT as part of an occupational health program.

2.4.2 Re-exposure is also a reality in this setting. We don’t fully understand how TPT may affect innate immunity, and this concerns some of the clinical staff. It is important to acknowledge these concerns which suggest that not all HWs will opt for TPT until better evidence is available.

We are aware of the concerns about the effect of TPT on innate immunity and specifically asked about this as part of our interview guide. However the participants in our interview cohort did not express this concern.

We also thought it was very important to report that HWs raised re-exposure as a concern, please see lines 732-736:

HWs desire to reduce their TB risk and willingness to consider TPT was tempered by the broader view of the high risk of re-exposure given current workplace constraints, which many HWs were concerned could compromise long-term treatment efficacy.

2.4.3 Similarly, side effects of the regimen are a concern, regardless of whether it is IPT, 3HP, or some other regimen. Nurses especially would be familiar with the hardships of side effects based on patient experience. Other research has suggested that this prevents health workers and patients from opting for prophylaxis, which may limit uptake.

- We have added text to address this in lines 731-733:

We note that HWs expressed concerns about side effects, which may have been impacted by seeing patients experience side effects of TB drugs. Knowledge of shorter regimens with better safety profiles was also limited.

2.4.4 It would also be helpful to readers from outside LMICs to provide some context around cost concerns. What does a TB ward look like? Are negative pressure rooms available? What other competing risks to patients and HWs are present? How might we go about advocating for systemic changes to prevent TB and other nosocomial infections in light of limited resources?

Cost concerns are undoubtedly a major challenge for TB care delivery as increased resources are critical to enable systemic changes to prevent TB and other nosocomial infections. We have revised and added to the text in lines 761-840 as follows, which includes the importance of aligning disease specific efforts within broader health systems strengthening initiatives:

We acknowledge the challenges of implementing LTBI testing and treatment in real world hospital settings in high incidence countries, where resources for implementing other TB infection control measures such as expanded triage, airborne isolation capacity, and ventilation, are limited. Nonetheless, we think there is an imperative to reduce the unacceptable risk of occupational TB faced by HWs, which can be mitigated by comprehensive approaches that include LTBI testing and treatment. Interventions should not focus on changing HW actions alone but must be coupled with administrative and infrastructural support. This could facilitate implementation of recommended TB infection control measures, in conjunction with LTBI testing and TPT as part of an occupational health programme, which HWs thought was the responsibility of the hospital to provide. Adopting a multi-pronged approach to addressing HW TB is necessary to prioritize the importance of TB transmission prevention. Incorporating TB infection control within broader health systems strengthening efforts can further these goals[46].

Minor comments:

2.5 Introduction:

There are a lot of acronyms introduced that may be unfamiliar to readers outside the field of TB. I would suggest spelling out infection control and Tygerberg hospital to reduce the number of acronyms.

We have spelled out Tygerberg hospital and TB infection control as suggested.

2.6 Line 94: Please reword this sentence, as hypothesis testing is not appropriate for qualitative data, as the findings are inductively derived.

We rephrased this as ‘We aimed to use’.

Methods:

2.7 Was a particular methodological orientation used in this study (e.g. ethnography, grounded theory)?

We used an inductive approach to analyzing our data and reference thematic analysis (see line 189).

2.8 Line 144: Who was the interview guide pilot tested on, and were any revisions made? It would be helpful to share the interview guide in a supplement. In particular it would be interesting to note in the findings whether any of the interviewees had previously had TB or lost a colleague to TB.

We revised the text in lines 174-175 as follows:

We piloted our interview guide with two research nurses and made revisions to ensure our questions were clearly and simply phrased.

2.9 Information on gender of participants should be included in methods or findings.

We included this in line 157 of the methods.

2.10 Lines 164: This section intersperses a lot of findings in the methods. Perhaps this detail would be better shared as a table or graph in the beginning of findings. I leave this to the authors to decide.

Thank you, we have revised this section extensively to clarify that this is part of the methods, please see lines 197-241.

2.1.1 Line 167: It would be helpful to add a line or two explaining a network approach for those who are unfamiliar.

We added the following text in lines 219-220:

..used a networked approach to consider different roles, resources and health systems factors related to occupational TB risk and risk reduction.

2.1.2 Lines 172-174: this sentence was a bit confusing and could benefit from rewording.

We revised this text in lines 220-226 as suggested:

Using risk, care, information, and protection as categories to analyze and organize the themes a second time, we described four revised domains that included 1) individual and systemic constraints and 2) actions that HWs and health systems took in anticipation and as a result of 3) HWs’ risk of occupational TB. Examining all references to constraint and action in a final level of data analysis we noted that a final domain, 4) desire, helped integrate the previous iterations of analysis.

2.1.3 Lines 184-85: Is it relevant that these data were analysed in parallel with SAFE study results? Did one inform the other?

We have removed this sentence.

Findings:

2.1.4 Lines 374-377: This sentence is long and difficult to follow. Please consider rewording.

We have revised this text in lines 616-619 for clarity:

Several HWs thought LTBI testing should be offered to staff given the potential benefits of HWs knowing whether they had been infected, which may lead to improved implementation of TB-IC measures. Others mentioned the importance of being able to provide effective treatment for a testing programme to be worthwhile.

Discussion:

2.1.5 Lines 438-439: I would advise rewording this sentence, as this is not the first study to look at perceptions of TPT in HWs in a high-incidence setting; however, the findings are very relevant.

We have revised this sentence in lines 689-691 as suggested:

Our study provides an in-depth examination of HWs’ perspectives on LTBI testing and treatment, in the context of unusually high occupational risk in a high TB incidence setting and global impetus for LTBI treatment programs.

2.1.6 Lines 499-501: This is a very important finding and a strength of your approach.

Thank you. 

Additional Editorial Comments addressed:

We have reviewed these and edited these accordingly.

2. Please include a copy of the interview guides used as a Supporting Information file.

We have included this as requested. 

The datasets generated during and analysed during the current study are not publicly

available since data consists of interview transcripts, which pose the risk that a person

could be recognized from their interview narrative and the description of their facility.

Any specific data requests can be made to the corresponding author, Dr. Nathavitharana, and/or the corresponding research ethics committee. This study was approved by the Stellenbosch University Health Research Ethics Committee, South Africa (Ref # N17/01/004, contact afortuin@sun.ac.za) and the Institutional Review Board of Brigham and Women’s Hospital, Boston, USA (Protocol #: 2017P000539, contact agargiulo1@bwh.harvard.edu). 

4. Please amend the manuscript submission data (via Edit Submission) to include authors Ananja van der Westhuizen, Helene-Mari van der Westhuizen, Hridesh Mishra, Annalean Sampson, Jack Meintjes, Edward Nardell, Andrew McDowell, Grant Theron.

We have done this.

---

## [Editor Report · Decision Letter 1]

23 Jun 2021

“If I've got latent TB, I would like to get rid of it”: Derivation of the CARD (Constraints, Actions, Risks, and Desires) Framework informed by South African healthcare worker perspectives on latent tuberculosis treatment

PONE-D-21-03741R1

Dear Dr. Nathavitharana,

We’re pleased to inform you that your manuscript has been judged scientifically suitable for publication and will be formally accepted for publication once it meets all outstanding technical requirements.

Kind regards,

Amrita Daftary

Academic Editor

PLOS ONE
---

## [Editor Report · Acceptance letter]

9 Aug 2021

PONE-D-21-03741R1 

*“If I've got latent TB, I would like to get rid of it”:* Derivation of the CARD (Constraints, Actions, Risks, and Desires) Framework informed by South African healthcare worker perspectives on latent tuberculosis treatment 

Dear Dr. Nathavitharana:

I'm pleased to inform you that your manuscript has been deemed suitable for publication in PLOS ONE. Congratulations! Your manuscript is now with our production department. 

Kind regards, 

on behalf of

Dr. Amrita Daftary 

Academic Editor

PLOS ONE